# Calibration of CO, NO_2_, and O_3_ Using Airify: A Low-Cost Sensor Cluster for Air Quality Monitoring

**DOI:** 10.3390/s21237977

**Published:** 2021-11-29

**Authors:** Marian-Emanuel Ionascu, Nuria Castell, Oana Boncalo, Philipp Schneider, Marius Darie, Marius Marcu

**Affiliations:** 1Faculty of Automatics and Computers, Politehnica University of Timisoara, 300223 Timisoara, Romania; oana.boncalo@cs.upt.ro (O.B.); marius.marcu@cs.upt.ro (M.M.); 2Norwegian Institute for Air Research (NILU), 2007 Kjeller, Norway; ncb@nilu.no (N.C.); ps@nilu.no (P.S.); 3National Institute for Research and Development in Mine Safety and Protection to Explosion–INSEMEX, 332047 Petrosani, Romania; marius.darie@insemex.ro

**Keywords:** air pollution sensors, air quality monitoring, data quality, electrochemical sensors, low-cost sensors, sensor calibration

## Abstract

During the last decade, extensive research has been carried out on the subject of low-cost sensor platforms for air quality monitoring. A key aspect when deploying such systems is the quality of the measured data. Calibration is especially important to improve the data quality of low-cost air monitoring devices. The measured data quality must comply with regulations issued by national or international authorities in order to be used for regulatory purposes. This work discusses the challenges and methods suitable for calibrating a low-cost sensor platform developed by our group, Airify, that has a unit cost five times less expensive than the state-of-the-art solutions (approximately €1000). The evaluated platform can integrate a wide variety of sensors capable of measuring up to 12 parameters, including the regulatory pollutants defined in the European Directive. In this work, we developed new calibration models (multivariate linear regression and random forest) and evaluated their effectiveness in meeting the data quality objective (DQO) for the following parameters: carbon monoxide (CO), ozone (O_3_), and nitrogen dioxide (NO_2_). The experimental results show that the proposed calibration managed an improvement of 12% for the CO and O_3_ gases and a similar accuracy for the NO_2_ gas compared to similar state-of-the-art studies. The evaluated parameters had different calibration accuracies due to the non-identical levels of gas concentration at which the sensors were exposed during the model’s training phase. After the calibration algorithms were applied to the evaluated platform, its performance met the DQO criteria despite the overall low price level of the platform.

## 1. Introduction

Air pollution is a major concern that affects not only the environment but also the health of individuals. Studies show that air pollution is responsible for many diseases, such as pulmonary and cardiovascular diseases, and that these conditions are only observable after a long exposure time [1]. Given its importance, the Environmental Department of the European Union released a European Air Quality Directive on ambient air quality and cleaner air for Europe [2] that sets the thresholds for environmental and health protection. The directive also sets the data quality objectives (DQO) requirements that air quality measurement data must meet after calibration. Low-cost sensor platforms, in order to be effectively used as indicative measurements in the EU, must meet the minimum requirements for expanded relative uncertainty with respect to reference monitoring stations defined in the directive [2]. Today, most cities have municipally managed reference stations to monitor air quality. Solutions relying solely on reference stations have an important shortcoming in the context of large-scale usage by individuals, which is limited spatial representation of the measured pollutant concentrations. Additionally, the high unit price of a reference station (€50–100.000) and the costs associated with its maintenance limit the number of instalments, yielding low spatial coverage. This is especially the case in small cities and less-developed economic regions.

All of these reasons, together with the rapid development in sensor technology, led researchers to build and use low-cost sensor platforms that would enable better spatial and temporal coverage [3]. Professional air monitoring stations typically measure pollutant levels of: carbon monoxide (CO), nitrogen oxides (NO_2_ and NO), particulate matter (PM_10_, PM_2.5_), ozone (O_3_), and environmental parameters (e.g., temperature and relative humidity). However, the use of low-cost sensors for air quality measurement faces several challenges, such as high uncertainty of the values, internal sensor errors, and poor selectivity of the target pollutant [4].

Low-cost air quality sensors employ one or several of the following measurement techniques: particulate matter sensors [5,6], electrochemical sensors [7,8,9,10,11,12], and metal-oxide sensors [9,11,13]. Irrespective of their type, all low-cost sensors are plagued by errors, yielding differences in measured data with respect to a high-quality reference. As argued in [4], there are two main sources of errors: internal errors, which are related to the sensor’s working principle, and external errors, which are caused by the working conditions. The most common internal sensor errors are: dynamic detection boundaries, non-linear response, and sensor drift. For external errors, the most representative is the low selectivity of the target gas, which is influenced by the cross-sensitivity with the other gases. Most of these errors can be mitigated using laboratory and colocation calibration methods.

In this study, we evaluated a low-cost air monitoring platform (Airify) measuring a wide variety of air parameters (i.e., CO, NO_2_, NO, O_3_, SO_2_, CO_2_, PM_10_, PM_2.5_, PM_1_, temperature, relative humidity, and atmospheric pressure), and we discuss the challenges of its calibration. Our group designed the proposed platform after analysing current solutions and developments in low-cost air quality monitoring devices. The sensors used in our devices were evaluated by other studies [10,14], and the reported results showed promising usability of them in a real-world scenario. The contribution of this work is the development and validation of calibration models based on state-of-the-art multivariate linear regression (MLR) models and random forest (RF) algorithms [5,8,14,15,16] such that the low-cost sensor platform meets the DQO.

## 2. Data Correction and Calibration Methods

The topic of low-cost air quality sensor calibration and correction algorithms has been the focus of scientists for more than a decade. New sensors emerging on the market, such as metal-oxide [9] and electrochemical [10] sensors, or those using light-scattering principles [5], offer new research opportunities. Recent works [3,15,17] target the collection of real-time air quality data with the aim of improving the existing infrastructure of fixed air quality monitoring stations in cities by improving temporal and spatial resolutions.

In order to make use of low-cost sensors, a couple of steps are required [7,8,9,10,14,15,17,18,19,20]:design a small device using low-cost sensors;calibrate the device in closed controlled chambers in a laboratory using known gas concentrations;assess the effectiveness of the calibration algorithm by placing the units outside in colocation with a reference station;compare the calibrated results with the reference results.

If the results are unsatisfactory with respect to DQO, the algorithm will be revised. Note that not all sensors can be successfully calibrated to a high-enough degree of confidence. Two important keys in choosing the candidate sensors for calibration are their levels of detection (LoD) and their cross-sensitivity with other pollutants. The lower LoD should be close enough to 0 or to the lower known atmospheric levels of the target pollutant, whereas the cross-sensitivity with other pollutants should be minimal or non-existent.

A typical method to assess the reliability and accuracy of low-cost sensors is to colocate them in the field against an official air quality station. Laboratory calibration is limited to reducing the dynamic boundaries of the internal errors (i.e., offset and gain errors [4]). The colocation might help mitigate the cross-sensitivity and the selectivity errors if the reference station is representative for the final place where the sensors will be deployed. After enough air quality data are collected, models can be developed to correct the data and calibrate devices in order to meet the regulatory standards.

The state-of-the-art calibration methods are summarized in Table 1. The table shows the state-of-the-art results, the underlying equation, and the predictors used for each gas. Two main categories of methods were identified: MLR with different regressions such as simple [7], multivariate [9,11,18], orthogonal [9,11,14,21], polynomial, and exponential curve fitting [16]. Other methods such as Gaussian processes [8,15], RF [21], and hybrid RF [8] algorithms are less commonly used.

Next, we briefly discuss the state-of-the-art calibration for each gas. The MLR calibration methods are the simplest models used to calibrate gas sensors, yielding good results [5,8,11,16]. For the MLR approach, each gas has its own equation. In this regard, for the CO and O_3_ gases, the best equations are those reported in [8]. According to [7,10], environmental changes are best accounted for using temperature and relative humidity measurements. O_3_ and NO_2_ gases have high cross-sensitivity, making the presence of sensors for both parameters mandatory [4].

For the NO_2_ gas, the information of the wind-speed and wind-direction meteorological data significantly improved the sensor response [16]. The results from [16] also suggest that there is a trade-off between the number of model parameters and the result accuracy. Note that meteorological data are generally not used as covariates for NO_2_ calibration [15,23]. We include this information in Table 1 since the model that reported the best performance according to state-of-the-art studies considered them.

In the last couple of years, many studies have tried to find nonparametric models in order to correct data from low-cost sensors [8,21]. This is prompted by the fact that sensor calibration is highly dependent on location. Therefore, once the devices are moved to another place, calibration is required [22]. The results suggest that the RF models work better for the NO_2_ and O_3_ gases, being the only algorithm that can correct both parameters with a factor of determination r^2^ > 0.7 [8].

The RF algorithms work better inside the training boundaries of the values but perform poorly once they encounter a value greater or smaller than what the algorithm has seen in the training dataset. To mitigate this shortcoming, the authors of [8] proposed a hybrid algorithm using the standard RF model when the new value is within 10% of the limits of the training dataset and an MLR model if the values are close to or outside the limits of the training dataset. Their results show that the hybrid RF approach performs better for CO and O_3_ than the classic RF model.

## 3. Implementation of Calibration Models

The calibration methods proposed in this work were designed for the air quality measurement device platform described in [24], under the premise that low-level data acquisition and filtering is available [25].

### 3.1. Devices and Platform Description

The air monitoring device, described in Figure 1, is equipped with electrochemical Alphasense (UK) A4-type sensors (CO, NO_2_, NO, O_3_, and SO_2_), which are the smallest and cheapest 4-electrode sensors designed for ppb levels; a non-disruptive infrared (NDIR) pyroelectric sensor for CO_2_; and a light-scattering sensor from PlanTower for PMs [25], which are placed in a closed case. The case is equipped with a fan and holes to allow good air flow over the sensors.

Compared to the state-of-the-art, this platform uses a diverse selection of the lowest-cost sensors. This diversity creates incentives for developing better calibration models, while at the same time limiting the cost per unit to approximately €1000. An overview of the device sensors used, and those from related work, is presented in Table 2. It is worth emphasizing that some research studies use sensor clusters [7,8,10], whereas others are limited to only a subset of sensors due to the manufacturer limitations (devices are off-the-shelf), or due to the rather high unit cost [5,9,16,18]. The majority of projects rely on electrochemical sensors from Alphasense B Series or Citytech. These sensors are expensive compared to those used in this work. Typically, the higher the price, the better the sensor sensitivity. The price and the sensor sensitivity are correlated. Thus, they can detect low ppb levels of a target gas, making the signal acquisition circuit less prone to noise.

In order to reduce this noise, we designed an in-house acquisition board made of reliable circuitry that is capable of assessing any variance and noise introduced by other electronics, such as wireless communication modules or fan start/stop periods [25].

### 3.2. Testing Environment

For the proposed calibration, we used five Airify units. Our devices were colocated together with an air quality monitoring reference station in a calibration facility for the duration of about one month (6 January 2021–2 February 2021) in the city of Petrosani area. The station is located outside, and its purpose is to continuously measure the air quality parameters for the city of Petrosani. The reference station uses sensors that are periodically calibrated and certified by the authorities. The reference sensor types and calibration certificate numbers are presented in Table 3.

The group of Airify devices under test were placed in a box as depicted in Figure 2a. To provide the same pollution gas to the devices, we inserted a hoe into the pipe used by the reference station. The air was then pumped inside the box using a 6 L/min pump, as can be observed in Figure 2b. Our units were equipped with new sensors for all parameters, and this study is the first exposure of them to outside conditions.

The box used had some shortcomings: the temperature might be higher inside the box, and it might lower the gas concentration sampled by the devices. These limitations are a consequence of box-made materials and their properties of thermal insulation and gas-absorbing capacity. The first shortcoming had a higher impact on our models because temperature was one of our predictors and is especially important for electrochemical sensors. Therefore, it is mandatory to use accurate temperature measurements during the training period. We mitigated this problem by calibrating the temperature reported by our devices using a simple linear regression between our measurements and the reference ones. The results using this approach had a coefficient of determination r^2^ = 0.99, which makes the calibrated value close enough to the value measured by the reference. These calibrated values were then used in the proposed calibration models. The box’s gas-absorbing property was taken into account by the method imposed to bring the gas around the devices. Using the pump, we ensured a constant flow of air around the devices, while the exhaust was carried out by leakage; thus, the capacity of the box to absorb pollutants was limited.

Measurements were performed using the internal sampling rate of both Airify devices and the reference station. The reference station had a sampling rate of 1 sample every 3 min, whereas our devices have a sampling rate of 1 sample per minute. Thus, the data series needed synchronisation during post-processing. Given our platform response time, the synchronisation choice was proven around 3 to 6 min.

In addition to this, we performed other post-processing computations in order to filter out outliers and erroneous values from the data. The first step was done in accordance with the lower and upper LoDs imposed by the reference station’s sensors. We removed any data point that fell below the detection limit of the reference station from the dataset. This led to the removal of a small percentage (1.5%) of NO_2_ data points. The lower LoD of all the reference sensors can be seen in the last column of Table 3. It is important to note that we did not remove any data that fell below the LoD of our device sensors. This is justified by the fact that we aimed to see if a calibration model can overcome the LoD of low-cost sensors and provide useful information about the measured phenomena. The LoD of our platform’s sensors are presented in Table 4. The results showed that the LoDs were similar for CO but were 30 times less for NO_2_ and O_3_. The second step was to remove the data points that were outside the equivalent of three times the standard deviation. We also considered them as outliers, and we removed the recordings from both the reference and the test data series.

The consequence of the short colocation period was the small value range reported by the reference for the NO_2_ parameter. Therefore, we did a small simulation for about 1.5 h at the end of the measurement period. For the last couple of colocation days, we used a mixture of diluted exhaust gas from a diesel engine (Euro 1) to make sure that we obtained CO and NO_2_ parameters above the LoD threshold. In this way, the sensors were exposed to a high diversity of pollution levels, since, in the monitoring period, the NO_2_ values were relatively low.

### 3.3. Calibration Models Description

Various calibration models have been used to test our devices and assess the effectiveness of the platform after calibration. Our approach uses a combination of parametric MLR models and RF non-parametric ones.

Linear regression models are the most simple and common approaches to calibrate air quality monitoring data [8]. These models use a linear function that maps the input features to an output value. Thus, we aimed to find a linear dependency between the measured data of the device and the reference data. Using a similar approach, we also implemented quadratic equations where we used powers of some input parameters such as temperature and relative humidity. In this regard, we proposed Equations (Equation 1)–(Equation 6) to calibrate the CO, NO_2_, and O_3_ data. Along with them, we also used the equations reported in Table 1 (lines 1, 2, and 3), since they are the best state-of-the-art results. For the NO_2_ equation from Table 1 (line 2), we disregarded the wind direction and the wind speed information for two reasons. First, our device platform does not support them for cost reasons. The meteorological station information has a low spatial coverage resolution (i.e., city level instead of street level, as required in our case). Second, this information is related to data prediction, rather than data calibration [23,26].

The models from Equations (Equation 2), (Equation 4), and (Equation 6) leveraged our array of sensors. We constructed the models using the Akaike information criterion (AIC) score [27] for each measured parameter in order to take into account only the meaningful predictors. Using the AIC score, we found that the measured PM_10_ was a strong predictor of CO and PM_2.5_ for the NO_2_ parameter, whereas NO had an influence on NO_2_ and O_3_. Incorporating these predictors into the model greatly improved the results.
(1)$$CO=\alpha_{0}s_{CO}+\beta_{0}$$
(2)$$CO=\alpha_{0}s_{CO}+\alpha_{1}T+\alpha_{2}s_{NO}+\alpha_{4}s_{PM_{2.5}}+\beta_{0}$$
(3)$$NO_{2}=\alpha_{0}s_{NO_{2}}+\alpha_{1}T+\alpha_{2}AH+\alpha_{3}s_{O_{3}}+\beta_{0}$$
(4)$$NO_{2}=\sum_{n=0}^{6}\alpha_{i}s_{i}+\beta_{0},where$$
$$s_{i}=\left[NO_{2},O_{3},T,CO,NO,PM_{10}\right]$$
(5)$$O_{3}=\alpha_{0}s_{NO_{2}}+\alpha_{1}s_{O_{3}}+\alpha_{2}s_{NO}+\beta_{0}$$
(6)$$O_{3}=\sum_{n=0}^{7}\alpha_{i}s_{i}+\beta_{0},where$$
$$s_{i}=\left[NO_{2},O_{3},T,RH,CO,NO,PM_{10},PM_{2.5}\right]$$

For all three gases under evaluation, we used the simple RF algorithm without the hybrid form, since both yielded similar results. The RF algorithm used was a random forest regressor that fits a number of classification decision trees on different subsamples of the original dataset. It uses averaging to control the overfitting problem. Using the sensor array, we proposed an RF algorithm composed of 300 decision trees with a maximum depth of 10 subnodes per tree. The proposed model used the mean square error as a fitting function in order to evaluate each decision split and was configured with a minimum of nine samples required to split an internal node and a maximum of nine features to be considered when searching for the best split.

## 4. Results and Discussion

For this study, we used five Airify units that were colocated with a reference station. Given that the devices performed similarly with negligible differences, hereafter we report the findings of a single device.

In the evaluation process, we used 3 min data average values of the device under test and of the reference high-quality station. For each model, we expressed the results as a coefficient of determination (r^2^), a root mean square error (RMSE), a normalised mean bias (NMB), and a normalised mean error (NME), and we computed the uncertainty of measurement using an orthogonal regression of the estimated values versus Reference [11]. The advantage of the RMSE metric is that it provides a combined measure of systematic and random error. The NME metric is more robust to outliers and cannot cancel negative and positive errors because both add to the sum. The NMB metric is a 0-mean metric showing the bias between the measured signal and the reference. If the NMB is negative or positive, the signal tends to underestimate or overestimate the reference. Using the normalised versions of these two metrics better presents the errors with respect to signal noise. For the RMSE, NMB, and NME metrics, smaller values indicate better agreement between the signals.

A certified calibration procedure should satisfy the EU DQO having a low RMSE and NME and a very-close-to-0 NMB. The accuracy of the data is also reflected in the measurement uncertainty metric (see [2] for a detailed description). In Table 5, we present the four classes of devices classified according to the relative uncertainty of the measurement. The relative uncertainty, U_r_, was calculated using Equation (Equation 7) with $b_{0}$ and $b_{1}$ being the slope and intercept of the orthogonal regression, respectively, whereas the sum of the square of residuals (RSS) was computed using Equation (Equation 8) (see [11] for further details).
(7)$$U_{r}\left(y_{i}\right)=2\left(\sqrt{\frac{RSS}{n-2}-u^{2}\left(x_{i}\right)+{\left[b_{0}+\left(b_{1}-1\right)x_{i}\right]}^{2}}\right)/y_{i}$$
(8)$$RSS=\sum {\left(y_{i}-b_{0}-b_{1}x_{i}\right)}^{2}$$

In Table 6, we present a summary of the proposed and evaluated models represented by the r^2^, RMSE, NMB, NME, slope, and intercept obtained for the validation period. The validation data used consist of 20% of the total samples selected randomly. This reduces the chance of the model being overfitted. The correlation between the predicted values of our models and the reference is presented as a scatter plot in Figure 3. The red line represents the one-to-one line between the reference values and a perfect calibration model, whereas the blue line is composed of the slope and intercept values of our model obtained during validation. Ideally, the two lines match. Their proximity is an indicator of the quality of the calibration model. The results show that CO presents a strong linear dependency for all models, whereas the NO_2_ and O_3_ values exhibit a higher variance mainly due to the fact that the 3 min data averaging was used for increased temporal granularity.

Using Equation (Equation 2) and the RF models, which take advantage of the wide variety of sensors on our devices, the CO concentration outperformed the state-of-the-art models in terms of r^2^. The CO improvement was due to the good correlation of our platform data and the measurement data from the reference station. Thus, good results in CO data measurement were obtained using the CO sensor alone. When using other sensors, a further 3% improvement was achieved. Compared to the results reported in the state-of-the-art, we obtained a 9% improvement in r^2^ for model 1, with respect to [8] and a 12% improvement when considering the RF model.

For the NO_2_ parameter, our results did not achieve the best of the state-of-the- art [16] (r^2^ of 0.5 versus 0.82), as a result of the fact that we did not include wind-speed and wind-direction information and because, in our colocation period, the values for the NO_2_ parameter were very low, making the sensor operate at its detection limit. Similar results were obtained using RF models with a maximum r^2^ of 0.65. Furthermore, this performance degradation can also be attributed to the fact that our sensors are one order of magnitude cheaper and smaller than the ones reported in [16]. During the evaluation period, NO_2_ concentration levels were low, and many values were below the limit of detection state by the sensor manufacturer; a good calibration procedure should assess this limitation. Using exhaust gases to increase pollution levels showed better correlation results since the values were well above the limit of detection.

Finally, the O_3_ sensor showed a very good correlation with the measured reference value. Thus, we obtained similar results as reported in [8] using a cheaper sensor in conjunction with the MLR models. For the nonparametric RF model, we obtained an improvement of 12% with respect to similar RF models from state-of-the-art studies and an overall improvement of 2% (r^2^ = 0.92) with respect to all the studies and projects investigated by us.

Based on Equation (Equation 7), the relative expanded uncertainty was computed for all three candidate sensors taking into account the reference uncertainty. Figure 4 presents the plot of U_r_ against the reference measured during the validation phase of all the models considered. We report the uncertainty with a coverage factor of 2, meaning that the level of confidence in the reported values was 95%. It is important to note that values below the DQO indicative level can be used as supplementary pollution information to reference measurements. Thus, devices that report values within this level of uncertainty have a high confidence in measuring the correct concentration of pollutants. Moreover, Figure 4 shows that the nonparametric RF models performed better, and the devices under test can reach the DQO indicative class for small ppb values of measured gases. The NO_2_ gas was the only candidate that only reached the DQO estimation class and not the indicative class. However, as uncertainty decreases with increasing measured gas concentration, we can infer that, for values greater than 100 ppb, our devices are able to reach the indicative class for this parameter as well. Such values are not uncommon in the peak traffic hours of many large and medium cities.

Figure 5 presents the target diagram for all candidates. This summary diagram highlights the relationship between the root mean square difference (RMSD) and the bias (B) for both the reference and the proposed model [28]. The distance between each point and the origin represents the total RMSD between the model and the reference. The diagram also shows whether the standard deviation of the model is larger (X > 0) or smaller (X < 0) than the reference. Figure 5 uses the bias and the RMSD normalised by the standard deviation of the reference. The CO sensor scenarios and the RF model for O_3_ yielded good results with points inside the circle radius of 0.5, whereas the NO_2_ and the other O_3_ models had slightly lower results since they were plotted in the circle radius of 0.7. The target diagram only included the cases having an RMSD smaller than one. It is worth noting that, without a calibration model, all sensor data except that of the CO sensor are unreliable.

For the proposed models, we obtained comparable and even better coefficients of determination for all three candidate gases with slopes very close to 1 and intercepts between 10 and 100. With respect to the DQO imposed by the EU, we concluded that all sensors can meet the estimation level (class 3) on intervals that can be found outdoors in real environments, and some models can reach the indicative level (class 2). The indicative level was only obtained for CO on all models evaluated and for O_3_ when the RF models were used; whereas, for NO_2_, it was only obtained when the RF models were used and for high concentrations of the parameter.

## 5. Conclusions

In this study, we developed and evaluated calibration models (MLR and RF) for a low-cost air quality platform for three gases: CO, NO_2_, and O_3_. Our platform, Airify, has a cost per unit that is five times less expensive than the state-of-the-art solutions and has a rich variety of sensors for air quality monitoring. The experiments showed that having a wide variety of available sensors yields better calibration models. Regarding related work, we obtained an improved coefficient of determination of 12% for the CO and O_3_ parameters, whereas, for NO_2_, we obtained similar results to other works, even without the use of external information such as wind direction and wind speed. NO_2_ sensor models did not exceed state-of-the-art results. However, Equation (Equation 4) and RF models with new predictors were able to improve the output calibration data.

For the CO gas, we obtained a high coefficient of determination and a low RMSD for the data even without any calibration. For the other gases, calibration models were mandatory. We evaluated our results with respect to state-of-the-art models by comparing the coefficient of determination, the RMSE, the relative uncertainty, and the target diagram.

The platform with the proposed calibration models meets the DQO for the estimation level for all of the validation concentration range, but only CO and O_3_ reached the indicative level requirements. NO_2_ values greater than 100 ppb can be observed during peak hours in urban areas and can also meet the indicative level of DQO, as the expanded uncertainty decreases with increasing concentration. The O_3_ sensor was able to meet the indicative DQO in the measured interval of values >20 ppb using the RF model.

It is important to note that by using the same calibration method (i.e., the RF model was used for all tree parameters with the same configuration), we obtained different results based on each individual parameter measured. This is the result of exposure to different levels of pollution concentration of each sensor type. The CO parameter constantly measured values >250 ppb (10 times its LoD), whereas the NO_2_ and O_3_ parameters were exposed to a concentration between 0–50 ppb, which was slightly above their LoD. Furthermore, it is well known that NO_2_ and O_3_ have a strong cross-sensitivity, making the results more susceptible to errors.

So far, our findings suggest that the Airify platform can be used in a real-world scenario as an indicative monitoring device for CO and O_3_ parameters. More evaluation is needed for longer periods of time for a definitive conclusion. This is a work in progress. The devices are in the process of undergoing colocated measurements, at which point we will re-evaluate the models on the newly collected data.

## Figures and Tables

**Figure 1 sensors-21-07977-f001:**
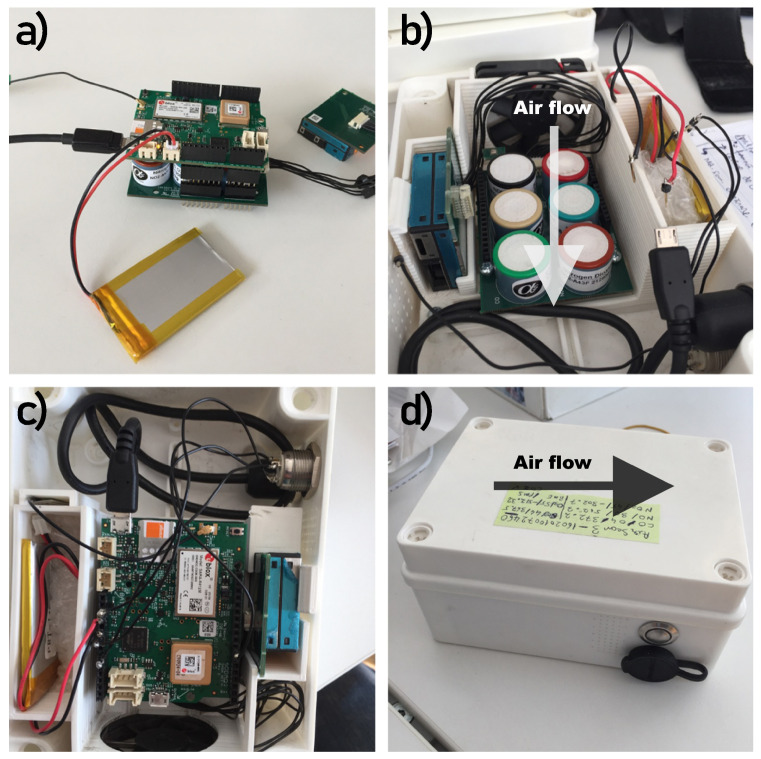
Airify air monitoring device: (**a**) the device without the case is made of two stacked boards: one for the sensors and another one used for processing purposes. The particle sensor is attached to the stack via a cable to reduce possible interference with its own fan. In case of a power shortage, the battery allows 4 h of autonomy. (**b**) The sensor board is placed at the bottom. A fan placed on the upper wall ensures that the air flows towards the sensors. The particle sensor has its own fan and the in/out openings are separated by the case to reduce the chance for a tunnel between them. (**c**) The processing board is placed on top. The two boards create a tunnel for air to flow over the sensors. (**d**) The Airify inside a case. The case has openings lengthways to create the air tunnel and also on the side for the particle sensor. On the side, the case has the power/indicator button and the charging port.

**Figure 2 sensors-21-07977-f002:**
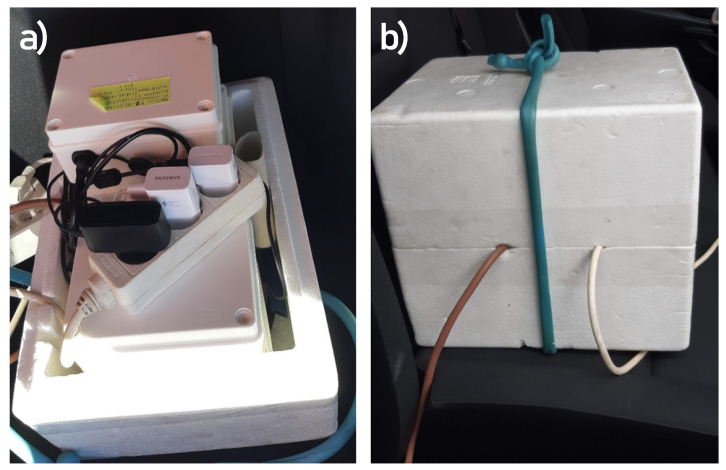
Devices under test placed in the exposure box: (**a**) inside the box, we placed 5 of our units with a power cord for each of them. (**b**) The box was closed, and the air was pumped inside the box via a 6 L/min pump to ensure a constant air flow. The air was taken from the same pipe used by the reference monitoring station.

**Figure 3 sensors-21-07977-f003:**
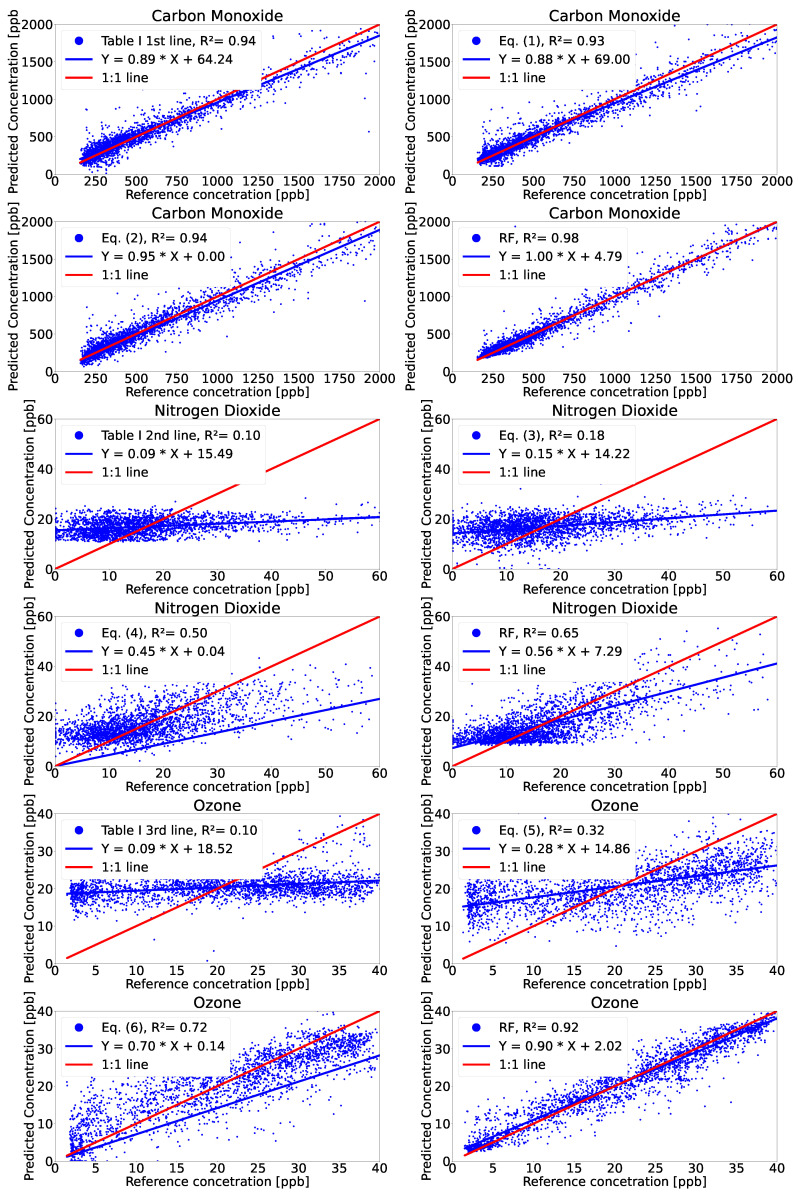
Scatterplot of calibration sensor data using the proposed models against reference measurements.

**Figure 4 sensors-21-07977-f004:**
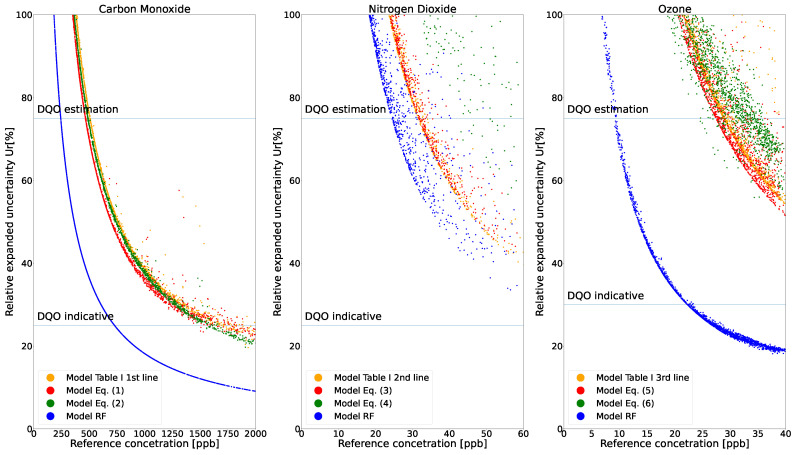
Relative uncertainty of the different calibration models versus reference measurements with a coverage factor of 2.

**Figure 5 sensors-21-07977-f005:**
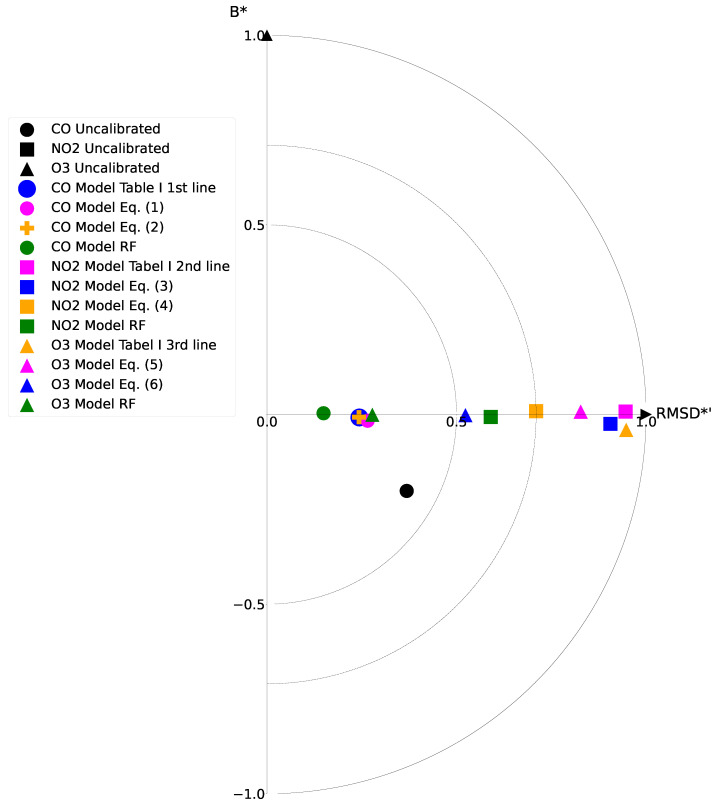
Target diagram for the proposed calibration models. For clarity, we present only the positive X of the target circle. The CO models except the RF one are overlapping around an RMDS*’ of 0.2 showing the good correlation between them. The uncalibrated NO_2_ and the uncalibrated O_3_ are outside the circle with radius 1 and are not present on the graph (nncalibrated NO_2_ B* = 4.14, RMSD*’ = 1.11; uncalibrated O_3_ B* = 14.32, RMSD*’ = 2.50).

**Table 1 sensors-21-07977-t001:** Overview of sensor calibration methods.

Calibration Method	Gas	Reference Papers	Results r^2^	Equation	Predictors
Linear regression	CO	[7,8,10,11,14,18,21,22]	0.85 [8]	CO = α_1_s_CO_ + α_2_s_CO_^2^ + α_3_T + α_4_T^2^ + α_5_RH + α_6_RH^2^ + α_7_s_CO_T + α_8_s_CO_RH + α_9_TRH + β_1_ [8]	CO, T, RH
	NO_2_	[7,8,11,14,15,16,18,21]	0.82 [16]	NO_2_ = β_0_ + β_1_$\sqrt{s_{{NO}_{2}}}$ + β_3_log(s_O_3__) + β_4_RH^2^ + β_5_T + β_6_$\sqrt{WS}$ + β_7_factor(WD) [16]	NO_2_, O_3_, RH, T, wind speed (WS), and wind direction (WD)
	O_3_	[7,8,11,14,15,18,21]	0.83 [8]	O_3_ = α_0_ + α_1_s_O_3__ + α_2_s_NO_2__ + α_3_T + α_4_RH [8]	O_3_, NO_2_, T, RH
Random forest	CO	[8,21]	0.77 [8]	Hybrid RF + LR on the edges [8]	-
Trees and hybrid	NO_2_	[8,21]	0.84 [8]	Random forest [21]	-
	O_3_	[8,21]	0.81 [8]	Hybrid RF + LR on the edges [8]	-

**Table 2 sensors-21-07977-t002:** Comparison of the device under test and literature devices.

Sensor Type	N. Castell et al. [7]	C. Malings et al. [8]	S. DeVito et al. [18]	L. Spinelle et al. [9,11]	V. van Zoest et al. [16]	O. A. M. Popoola et al. [10]	M. H. Bergin et al. [5]	Current Work
CO	Alphasense CO-B4	Alphasense CO-B4	-	MICS-4514, TGS-5042	-	CO-AF, CO-B4	-	Alphasense CO-A4
CO_2_	-	NDIR SST Sensing	-	Gascard NG, ELT Sensors S-100	-	-	-	Alphasense NDIR
NO	Alphasense NO-B4	Alphasense NO-B4	Alphasense NO-B4	Citytech NO-3E100	-	NO-A1, NO-B4	-	Alphasense NO-A4
NO_2_	Alphasense NO2-B42F	Alphasense NO2-B42F	Alphasense NO2-B42F	Alphasense NO2-B4, Citytech NO2-3E50, MICS-2710	Citytech Sensoric NO2 3E50 ECN	NO2 A1	-	Alphasense NO2-A43F
O_3_	Alphasense OX-B421	Alphasense OX-B421	Alphasense OX-B421	Alphasense O3-B4, Citytech O3-3E1F	E2V MICS 2610	-	-	Alphasense OX-A431
SO_2_	-	Alphasense SO2-B4	-	-	-	Alphasense SO2-B	-	Alphasense SO2-A4
PM_1_	-	-	-	-	-	-	-	PMSA003
PM_2.5_	AQMesh	-	-	-	-	-	PMS3003	PMSA003
PM_10_	AQMesh	-	-	-	-	-	PMS3003	PMSA003
Temperature	Yes	Yes	Yes	Yes	Yes	Yes	Yes	Yes
Relative humidity	Yes	Yes	Yes	Yes	Yes	Yes	Yes	Yes
Sampling time	15 min	15 min	1 min	1 h	10 min	5 s	1 min	3 min

**Table 3 sensors-21-07977-t003:** Reference station sensor certificates.

Name/Type	Gas	Series	Lower Limit of Detection (ppb)	Uncertainty
Analyzer CO model APMA 370 (certificate 4393/31.10.2019)	CO	MB733KF9	50	3%
Analyzer model Serinus 40 (certificate G184-2020/ 23.09.2020/NO)	NO, NO_2_	15-0619	0.4	14.5%
Analyzer O_3_ model APOA 370 (certificate 40-2020/31.01.2020)	O_3_	TJRRSS70	0.5	2.5%

**Table 4 sensors-21-07977-t004:** Device under test lower limits of detection.

Manufacturer	Gas	Type	Lower Limit Of Detection (ppb)
AlphaSense	CO	CO-A4	20
AlphaSense	NO_2_	NO2-A43F	15
AlphaSense	O_3_	OX-A431	15

**Table 5 sensors-21-07977-t005:** Data quality objectives (DQO) of the European Directive [2].

Class	O_3_	CO, NO_2_
DQO reference measurements	Uncertainty = 15%	Uncertainty = 15%
DQO indicative measurements	Uncertainty = 30%	Uncertainty = 25%
DQO Objective estimation	Uncertainty = 75%	Uncertainty = 75%
Additional class	Uncertainty = 200%	Uncertainty = 200%

**Table 6 sensors-21-07977-t006:** Evaluated model prediction results.

Sensor	r^2^	Slope	Intercept	RMSE	NMB	NME
CO(MLR Model Table 1 (1st line))	0.94	0.89	64.24	155.07	−0.02	0.14
CO(MLR Model Equation (Equation 1))	0.92	0.88	69.00	153.44	−0.03	0.12
CO(MLR Model Equation (Equation 2))	0.94	0.95	00.00	150.90	−0.03	0.18
CO(RF Model)	0.98	01.00	4.79	30.59	−0.01	0.08
NO_2_(MLR Model Table 1 (2nd line))	0.10	0.09	15.49	10.43	−0.04	0.45
NO_2_(MLR Model Equation (Equation 3))	0.18	0.15	14.22	10.50	−0.05	0.45
NO_2_(MLR Model Equation (Equation 4))	0.50	0.45	0.04	8.57	−0.04	0.37
NO_2_(RF Model)	0.65	0.56	7.29	7.52	−0.03	0.32
O_3_(MLR Model Table 1 (3rd line))	0.10	0.09	18.52	10.20	−0.04	0.42
O_3_(MLR Model Equation (Equation 5))	0.32	0.28	14.86	9.11	−0.03	0.36
O_3_(MLR Model Equation (Equation 6))	0.72	0.70	0.14	5.82	−0.02	0.23
O_3_(RF Model)	0.92	0.90	2.02	3.14	−0.00	0.10

## Data Availability

The data presented in this study are available on request by the corresponding author.

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
