# Peer review of "Calibration of CO, NO_2_, and O_3_ Using Airify: A Low-Cost Sensor Cluster for Air Quality Monitoring"

_sensors, 2021, doi:10.3390/s21237977_

Round 1
Reviewer 1 Report
In figures 1 and 2 more explanations would be useful, possibly in the figure, but at least in the caption, to explain and highlight the various parts of the device.
Figure 5 is not understandable: perhaps the size of the individual points and the position of the table with references need to be reviewed.
Author Response
Point 1: In figures 1 and 2 more explanations would be useful, possibly in the figure, but at least in the caption, to explain and highlight the various parts of the device.
Response 1: Thank you for the input. We have addressed this remark in the caption of the figures. For clarity we added labels to each individual subfigure and described each of them individually. The new figures 1 and 2 are present in the revised paper.
Point 2: Figure 5 is not understandable: perhaps the size of the individual points and the position of the table with references need to be reviewed.
Response 2: Thank you very much for the input. We have addressed this issue and the new Figure 5 is present in the revised paper.

Reviewer 2 Report
The authors addressed well in constructing new model to satisfy DQO for economic sensors by modifying equations. However, overall experimental set-up for data measurement is required to be explained to exclude unexpected side effects or errors.
Provide the specification (limit of detection, manufacturer, and etc.) of each sensor used in this study by summarizing in the table for more and clear information.
Most low-cost sensors, based on metal oxide semiconductor type, are required to be fully recovered to the initial status due to its intrinsic gas absorption/desorption phenomena after several times of experiments.
In addition, there were not enough explanations how specific concentration of target gas was prepared and how target concentration was maintained inside of chamber.
Please explain the procedure in more details for readers to understand. As you know, it is very important to set up the precise experimental conditions in measuring data for calibration.
The apparatus in figure 2, closed box chamber was consisted of Styrofoam box, which generally is used for thermal insulation material, and power adapters It appears that inside temperatures are bit higher than the ambient temperature due to the heat possibly emitted from power adapters installed together and thermal insulation. Inside of the box. So, temperature of box and air flow might influence the results. In addition, Styrofoam may work as gas absorbing medium and affect gas concentration in closed volume, even it was just used for closed box. It is my suggestion to design an apparatus with other materials such as vacuum metal parts
Please explain the temperature of inside box and air flow drawings in testing environment section.
In results chapter, suggested model showed different correlation results depending on type of gases. Please explain why. There is only explanation about difference in results, not enough scientific consideration to support the difference
Author Response
Point 1: Provide the specification (limit of detection, manufacturer, and etc.) of each sensor used in this study by summarizing in the table for more and clear information.
Response 1: Thank you for pointing this out. We have enhanced Table 4 (which is now Table 3) where we presented the reference station individual sensors with the information about their limit of detection (LoD). Also, we added our own sensors with their LoD in the new Table 4. We do removed the data below the LoD of reference station, but we kept all of our device measurements regardless of their LoD as long as the reference was above its corresponding level since we wanted to find out if our device with the proposed calibration methods can provide sufficient accurate results. Moreover, we rewrote the Testing Environment subchapter adding this information more clearly between lines : 155-161.
Point 2: Most low-cost sensors, based on metal oxide semiconductor type, are required to be fully recovered to the initial status due to its intrinsic gas absorption/desorption phenomena after several times of experiments.
Response 2: Thank you for your remark, we share your opinion. We are aware of this limitation of metal oxide semiconductor sensors. We did not used this type of sensors in our experiment, but electrochemical ones and all of them were just opened from their case without prior exposure to any pollutants. The devices are still under testing condition at the moment for more laboratory and in the filed measurements. Further results will be sent to publishing in the near future. We also specify this on times 162-167.
Point 3: In addition, there were not enough explanations how specific concentration of target gas was prepared and how target concentration was maintained inside of chamber.
Please explain the procedure in more details for readers to understand. As you know, it is very important to set up the precise experimental conditions in measuring data for calibration.
Response 3: Thank you for your comment! We did our experiment using colocation of our units with a reference station and not a laboratory calibration. The concentration of a target gas is not known a priori since the reference station measure the air around it. Thus inside the chamber where we placed our devices the same air was taken from the pipe of the reference and was pumped inside the chamber using a 6 l/min pump. We rewrote the Testing Subchapter and this information was explained from line: 155-167.
Point 4: The apparatus in figure 2, closed box chamber was consisted of Styrofoam box, which generally is used for thermal insulation material, and power adapters It appears that inside temperatures are bit higher than the ambient temperature due to the heat possibly emitted from power adapters installed together and thermal insulation. Inside of the box. So, temperature of box and air flow might influence the results. In addition, Styrofoam may work as gas absorbing medium and affect gas concentration in closed volume, even it was just used for closed box. It is my suggestion to design an apparatus with other materials such as vacuum metal parts
Response 4: Thank you for your valuable information. We had also our concerns about the box chamber used in this study, but please note that we did not had control over the procedure as this part was undertaken by our colleague, PhD Marius Darie, at INSEMEX institute using their equipment. We also wanted to use a metal case for this procedure or at least to use the devices only with batteries but in the end the measurement was done as described above and more clearly in the revised paper. As for the fact that the box might work as gas absorbing medium, we cannot do much since as stated we did not have direct control over the measurement procedure. We do take into account the temperature as described in the remark point 6 below. All of this information was very valuable to us, and we include it in the revised paper on lines: 164-184
Point 5: Please explain the temperature of inside box and air flow drawings in testing environment section.
Response 5: As described on the above remark we did considered the difference in temperature between our device and the reference one. In this regard, we used the reference temperature in a simple linear regression function to correct the temperature measured by our device. Since the obtained values where very close to reference (factor of determination of 0.99) we did not explain this in the first version of the paper. We also had five of our units there for testing and we observed variations of temperature, therefore the temperature calibration was also done before the values were used in calibrating the proposed sensors. Based on your remarks it is clear that this is an important topic and should have been addressed from the beginning. We have made this information available in testing subchapter on lines 164-184
Point 6: In results chapter, suggested model showed different correlation results depending on type of gases. Please explain why. There is only explanation about difference in results, not enough scientific consideration to support the difference
Response 6: Thank you for pointing this out. We missed some explanations about this topic. We have only one model that is common to all three pollutants, the Random Forest one. Even if the RF results were better we had different results mostly because of he concentration level of each pollutant that our device measured and because of different level of cross-sensitivities between them. Firstly, the CO concentration level that was measured was always over 250 ppb and the CO-A4 sensor used has a LoD of 20 ppb, making the measurement range at least 10 times bigger. On the other hand, NO2 and O3 levels detected was in range o 0-50 ppb whereas the LoD for these sensors are 15ppb. Thus the errors on CO should be smaller than on the ones obtained for NO2 and O3 because the CO operates well over its LoD. Secondly, there is a very well known cross-sensitivity problem between NO2 and O3. This problem makes the detection of independent NO2 and O3 with low-cost sensors harder and thus the errors should increase. All of this explanation was also added in the paper on lines 359-361. Thank you again for your input on this. We overlooked this explanation.

Reviewer 3 Report
In the paper, Ionascu et al. present a low cost sensor array to monitor CO, O3, and NO2. The aim of the research is to calibrate the system to meet the EU quality objectives. This is a beneficial improvement because it can lower the costs for a wider monitoring system. The authors use various approaches and as the result they have succeeded to meet the target performance objectives within a certain concentration range for the compounds in question, in some cases requiring prior calibration.
The topic is interesting and relevant to the journal Sensors. However, I believe the paper has to be significantly improved before it could be considered for publication. Detailed comments in the following.
- Language needs to be checked, there are typos and grammar mistakes throughout the paper.
- In the abstract and the introduction, you should explicitly mention that you developed the Airify system (at least I assume you did after reading the manuscript). Otherwise, this is confusing to the reader.
- 78-84: All this is assuming that the sensors are good enough for the task. If the sensors are inadequate, even top calibration will not help. Perhaps comment on that.
- Table 1 is hard to read because of the formatting. I suggest including horizontal lines or blank lines or something. O3 needs to use subscript at one point.
- 101-102: You say that wind data are generally not used yet you show the formula. Confusing to the reader, provide more context for clarity.
- Testing section. What are five units? Five independent systems? Four systems plus calibration?
- 6 January to 2 February is not one month but about a month (technical comment). Also, 2021?
- It is not clear how you did the testing. Did you sample the outside air with all pollutants included (I suspect this is the case) or you simulated different concentration using some setup? The part with exhaust gas, this is simulation.
- I have problems with understanding Fig3. It is referred to only later in the paper. What are the 1-4 plots for each compound? Four parallels or four ways to extract the data from the measurements (It seems the last one). There are so many points that the blue and black lines are sometimes not even seen. Try making them bolder.
- I’d like to see some more discussion about the RF models you used. Did you use RF classification (assigning a data point to a bin) or RF regressor (using the RF to choose the best regression model)? Probably the second one but it should be explained better.
- One more conceptual question. You are always using linear regression. What about a polynomial function instead, would that improve the data? Perhaps it could work for NO2, looking at the data.
- Table 4, mention what is U (uncertainty)
- Fig4, the legend hides part of the data, which could have been avoided. The dots in the legends are so tiny that they are not possible to interpret in a printed version (also, I printed it b/w, so it is nearly impossible). Define what is coverage factor.
- In the text, more directly mention that the values under the lines are good, above the line are bad. It will help the reader who is not familiar with these regulation metrics.
- Figure 5, try to move the legend in a way that it does not overlap the target. Or at least make it clear that it does not overlap any of the data. Again, I am having problems with distinguishing points in a b/w print.
- I am missing some conclusion at the end, regarding the usability of the findings. You discuss the performances for each of the three compounds, but what about overall? For monitoring a city where the values are expected within certain ranges? Would you need to improve the system further or do you consider it good enough for practical use?
Author Response
Point 1: Language needs to be checked, there are typos and grammar mistakes throughout the paper.
Response 1: Thank you for the comment! We have read again the paper and corrected most of the errors we could find.
Point 2: In the abstract and the introduction, you should explicitly mention that you developed the Airify system (at least I assume you did after reading the manuscript). Otherwise, this is confusing to the reader.
Response 2: The Airify system was indeed designed by our group. We have studied the air quality solutions and sensors for the last five years and we experiment with a variety of sensor types for both small portable devices and fixed bigger ones. Currently we are finishing our wearable solution which will undergo calibration in the next couple of months and we also continued the tests of our units present in this paper. In the next period we will report our calibration performance over the winter, hopefully with results similar to the ones reported. As per your suggestion we added the fact that Airify is our own solution briefly on line 7 and also on lines 59-62.
Point 3: 78-84: All this is assuming that the sensors are good enough for the task. If the sensors are inadequate, even top calibration will not help. Perhaps comment on that.
Response 3: Indeed, it is important that the sensors used to have some degree of confidence. From our experience we could not used with success the metal-oxide ones (even if their price is lower and some studies reported good results), but using A4 series from AlphaSense in general we got decent results. Those sensors are also used with success in some studies we researched and we are confident that they can be calibrated to DQO indicative level. We added a comment on that on lines 82-87 as you suggested because our steps were too general making the non domain readers to believe all the sensors can be used as described.
Point 4: Table 1 is hard to read because of the formatting. I suggest including horizontal lines or blank lines or something. O3 needs to use subscript at one point.
Response 4: Thank you for pointing this out. We reformat the table and correct the indentation.
Point 5: 101-102: You say that wind data are generally not used yet you show the formula. Confusing to the reader, provide more context for clarity.
Response 5: Thank you for letting us know. In Table 1 we considered the best models (as fas as we researched) reported by state of the art studies. Usually, these predictors are not used in calibration since any parameter that is not directly measured by a device should not be used for calibration purposes (according to Hagler, G.S. et. al, reference no. 23 in the paper). van Zoest et. al. (ref. No. 15 in the paper) considered this information and obtained the best results we also included wind data in Table 1. We added more explanations on lines 112-113 as your suggestion to add more clarity.
Point 6: Testing section. What are five units? Five independent systems? Four systems plus calibration?
Response 6: Thank you for pointing this out. We had in colocation five of our Airify systems near a reference one. We did improve the Testing Environment subchapter. This particular clarification is present on lines 155-161.
Point 7: 6 January to 2 February is not one month but about a month (technical comment). Also, 2021?
Response 7: Thank you for this input. You are completely right and we did the modification as suggested on line 157.
Point 8: It is not clear how you did the testing. Did you sample the outside air with all pollutants included (I suspect this is the case) or you simulated different concentration using some setup? The part with exhaust gas, this is simulation.
Response 8: Thank you for this comment. Indeed the testing part should have been clearer. We rewrote it entirely, the modification for this particular comment can be found on lines 155-161.
Point 9: I have problems with understanding Fig3. It is referred to only later in the paper. What are the 1-4 plots for each compound? Four parallels or four ways to extract the data from the measurements (It seems the last one). There are so many points that the blue and black lines are sometimes not even seen. Try making them bolder.
Response 9: Indeed the 1-4 plots per each parameter represent four ways to extract the data from the measurements, one for each of the imposed models. Basically in each image we present the calibrated test values used to evaluate the model in cause where we also placed the 1:1 line as reference. The image was not very clear and we improved it by placing the images in a 2x6 format and by making the lines and legend bigger. It is to note that the new images present slightly different values since we took the testing data randomly from the evaluation samples and we regenerate the image. We also reposition the image closer to its reference. We also comment more on the image including your suggestions on lines 269-273.
Point 10: I’d like to see some more discussion about the RF models you used. Did you use RF classification (assigning a data point to a bin) or RF regressor (using the RF to choose the best regression model)? Probably the second one but it should be explained better.
Response 10: Indeed we used the RF regressor for this study. We added more details as per your review on lines 233-240. Thank you for your input!
Point 11: One more conceptual question. You are always using linear regression. What about a polynomial function instead, would that improve the data? Perhaps it could work for NO2, looking at the data.
Response 11: Thank you for your great input. We did consider the quadratic equation for NO2 but the results were very similar with that we already obtained. We had a lot of analysis of NO2 values since in other tests we obtained better results but there the reference was not so reliable. It may be a problem of data synchronisation since we had 5 units and the results were similar between all of them, including poor correlation of NO2 values. In our latest tests (in laboratory calibration) we obtain better results for NO2 but there the environment was enclosed and controllable. This new tests will be publish at a later date since for the moment the units are undergo another colocation measurement at Norwegian Institute for Air Research (NILU) and we will evaluate also the long term calibration there. For this study we only introduced new models that on short term seems reliable giving us promising results for a real world scenario.
Point 12: Table 4, mention what is U (uncertainty)
Response 12: Thank you, we did the modification. Please note that Table 4 is now Table 5.
Point 13: Fig4, the legend hides part of the data, which could have been avoided. The dots in the legends are so tiny that they are not possible to interpret in a printed version (also, I printed it b/w, so it is nearly impossible). Define what is coverage factor.
Response 13: You are completely right, we did consider the Figure clear enough but looking again it is not so. We enlarged the dots in the legend and we modified the texts and legend positions. We checked the image on b/w and there is a greater difference between plots. Also, we explained the coverage factor of on lines 304-308.
Point 14: In the text, more directly mention that the values under the lines are good, above the line are bad. It will help the reader who is not familiar with these regulation metrics.
Response 14: Great remark! We did modify this as per your suggestion on lines 304-308.
Point 15: Figure 5, try to move the legend in a way that it does not overlap the target. Or at least make it clear that it does not overlap any of the data. Again, I am having problems with distinguishing points in a b/w print.
Response 15: You are right, this was very hard to realised with 12 different colours and markers, but we redone it in a clearer way. We added in the caption explanations about the overlapping dots.
Point 16: I am missing some conclusion at the end, regarding the usability of the findings. You discuss the performances for each of the three compounds, but what about overall? For monitoring a city where the values are expected within certain ranges? Would you need to improve the system further or do you consider it good enough for practical use?
Response 16: This is indeed a very important conclusion that we definitely missed. Thank you for pointing this out. We concentrate the conclusions only on current paper results. Definitely the results reported here are very encouraging and overall we do believe the system is almost ready to measure with high confident the air quality inside a city. At this point it is important to continue the analysis over a larger period of time, therefore in this moment our units at at Norwegian Institute for Air Research (NILU) where they undergo a colocation measurement for part of this autumn and past the winter. We are confident that the models will perform well, but without further data we only can say that they are good enough for this testing period. We added this information on lines 367-371. This study is a preliminary evaluation of those proposed models. We found this models by analysing the state of the art, then we employ maybe the lowest cost sensors that can do a decent job at detecting low level pollutant and we tune our previous prototype to this system. We do believe that having the entire array of pollutants (imposed by Environmental Protection Agency in the US and EU environmental agency) can help the models, and also we do believe that having a mass of low cost devices is the best way to map a city pollution that is meaningful for the individual. Thus, it is mandatory to create a device that is really affordable for end uses as well as for municipalities (especially in low income countries like Romania), and that can be calibrated to a degree that the reported values are reporting the reality. Otherwise, we will have a small amount of devices that measure locally and sometimes are well controlled by authorities.

Round 2
Reviewer 2 Report
It is suggestd that authors need to reflect revised discussions on your propsed model about different results regarding type of gas in between line 13 and 15 of ABSTRACT.
Still, some mispell and mistakes( ex. in line 371) are found. Please double check before final submission of revised version of draft.
Author Response
Point 1: It is suggestd that authors need to reflect revised discussions on your propsed model about different results regarding type of gas in between line 13 and 15 of ABSTRACT.
Response 1: Thank you! We have modified the Abstract accordingly with you comment.
Point 2: Still, some mispell and mistakes( ex. in line 371) are found. Please double check before final submission of revised version of draft.
Response 2: Thank you for the comment! We have read again the paper and corrected most of the errors we could find. We also removed the duplicate word as you suggested.

Reviewer 3 Report
The authors have addressed my comments. The paper now reads clearer and the ideas are presented better. Clearly, as this is work in progress, the impact of the results is not very high yet, but it is going in the right direction and I am sure the finalized setup will be useful.
The language still needs some fixes, such as the use of present perfect is not encouraged in scientific papers. I don't need to see the final revision, I trust the editor on this one.
Author Response
Point 1: The language still needs some fixes, such as the use of present perfect is not encouraged in scientific papers. I don't need to see the final revision, I trust the editor on this one.
Response 1: Thank you very much for the overall review of the paper! We tried to address most of the errors, and removed the present perfect on some part of the paper. It should read better now.
